# A Step Forward for Smart Clothes: Printed Fabric-Based Hybrid Electronics for Wearable Health Monitoring

**DOI:** 10.3390/s24216991

**Published:** 2024-10-30

**Authors:** Huating Tu, Zhenglin Li, Zihao Chen, Yang Gao, Fuzhen Xuan

**Affiliations:** 1College of Medical Instruments, Shanghai University of Medicine & Health Sciences, Shanghai 201318, China; tuht@sumhs.edu.cn (H.T.);; 2School of Mechanical and Power Engineering, East China University of Science and Technology, Shanghai 200237, China

**Keywords:** micro-electronic printing, ECG amplification circuits, textile electronics, smart clothes

## Abstract

Smart clothes equipped with flexible sensing systems provide a comfortable means to track health status in real time. Although these sensors are flexible and small, the core signal-processing units still rely on a conventional printed circuit board (PCB), making current health-monitoring devices bulky and inconvenient to wear. In this study, a printed fabric-based hybrid circuit was designed and prepared—with a series of characteristics, such as surface/sectional morphology, electrical properties, and stability—to study its reliability. Furthermore, to verify the function of the fabric-based circuit, simulations and measurements of the circuit, as well as the collection and processing of a normal adult’s electrophysiological signals, were conducted. Under 10,000 stretching and bending cycles with a certain elongation and bending angle, the resistance remained 0.27 Ω/cm and 0.64 Ω/cm, respectively, demonstrating excellent conductivity and reliability. Additionally, the results of the simulation and experiment showed that the circuit can successfully amplify weak electrocardiogram (ECG) signals with a magnification of 1600 times with environmental filtering and 50 Hz of industrial frequency interference. This technology can monitor human electrophysiological signals, such as ECGs, electromyograms (EMGs), and joint motion, providing valuable practical guidance for the unobtrusive monitoring of smart clothes.

## 1. Introduction

Smart clothing is an integration of textile materials and electronic information technology that can detect changes in the external environment or internal state and provide real-time feedback or interactions [1,2,3]. Sensors of smart clothing have gained a lot of attention because they are flexible, lightweight, and better wrap around the body to provide real-time and non-invasive health monitoring, including monitoring electrophysiology, chemical composition, and exercise motions [4,5,6]. As a matter of fact, it is a challenge to achieve truly unobtrusive monitoring [7]. Although the sensing unit was designed to be flexible and miniaturized, it still requires a large peripheral volume of signal computing equipment to process sensed signals, such as Arduino hardware [8] to produce electrocardiographs [9,10,11]. Additionally, standard PCB preparation equipment is expensive, and the process involves corrosive chemicals and other forms of environmentally damaging products [12,13]. The creation of integrated flexible electronics for information sensing, storage, and processing in wearable health monitoring devices is a critical issue.

Flexible printed electronics are additive manufacturing technologies, which have been proven to enable the full flexibilization of functional electrical circuits for wearable devices [14,15,16]. Chang et al. [17] utilized printing techniques to fabricate resistors, capacitors, and inductors on flexible plastic films and combined them to realize functional operational amplifiers and 4-bit digital-to-analog converters (DACs). However, that circuit had insufficient precision and arithmetic competence to handle the complex processing of biomedical signals. According to Yasser Khan et al. [18], while conductive inks have been widely employed in flexible sensing and displays, complicated data processing and communications continue to require silicon integrated circuits, such as functional modules of amplifiers, filters, analog-to-digital converters, and so on. Their team proposed a multi-substrate approach [19] in which hard and soft devices are built or constructed on different substrates and then joined or interfaced via connections. Additionally, Gao et al. [20] developed a smart wristband for real-time perspiration analysis, which integrated a multiplexed sweat sensor array on polyethylene terephthalate (PET) and a wireless flexible printed circuit board (FPCB). Although the above methods enable flexible and wearable applications, the key signal conditioning, processing, and wireless transmission operations were still conventional PCB circuits. Obviously, this reduces the flexibility and increases the risk of interconnection problems.

As an alternative, flexible hybrid electronics (FHEs) effectively solve the issue of flexible and wearable health monitoring devices by utilizing flexible printing technology to construct soft bio interfaces and peripheral circuits while simultaneously integrating silicon chips in a reliable manner. In 2018, the flexible electronics manufacturing company NextFlex created a flexible prototype based on the Arduino Mini, demonstrating the viability of flexible hybrid electronic systems for the first time. Subsequently, more and more 3D, stretchable electronics have been fabricated using multilayered sequential casting of elastomeric films such as polydimethylsiloxane (PDMS) [21,22].

Although the above strategies solved the issue of flexibilization in core arithmetic circuits, the issue still persists in external devices for smart clothing. To solve the issue in daily, wearable clothing, a series of attempts have been made by researchers to create flexible, fabric-based hybrid electronics. Zhuang et al. [23] utilized three-dimensional patterned technology to fabricate multilayered liquid metal circuits, combining high-density inorganic electronic components and organic stretchable fibrous substrates to create wireless, battery-powered, and battery-free bio-electronic systems attached to the skin. The approach is mostly used for stretching electronic skin, and it is complicated and expensive in terms of materials, equipment, and processes. Ma et al. [24] employed a practical chemical process of polymer-assisted metal deposition (PAMD) to achieved fabric surface circuit metallization and developed fabric-based wireless epidermal biosensing. However, this was not applicable to the creation of intricate, multilayer, non-crossing circuits, such as those with a through-hole, via-hole, and insulation between signal and ground layers.

To overcome the above problems, this research utilizes micro-electronic printing process on ordinary fabric to build multilayer hybrid electronics for wearable health monitoring (as illustrated in Figure 1). A series of characterizations such as those of the surface/sectional morphology, electrical properties, and stability were conducted to verify the reliability of the printed circuit. Additionally, simulations and measurements were performed to verify the feasibility of the signal processing and computation of the printed circuit. Furthermore, the electrophysiological signals of the ECG, EMG, and joint motions were collected and processed with an adult.

The main amplification and filtering functions of electrocardiographs were realized in this study by printing flexible hybrid circuits on fabric and can be worn comfortably and seamlessly integrate onto any part of the human body. It provides a solution of comfortability and the planar integration of key hardware circuits for unobtrusive monitoring, enabling a bold exploration for smart clothes.

## 2. Materials and Methods

### 2.1. Materials

Conductive circuit preparation by printing conductive ink on fabric is an economical, efficient, and reliable way to prepare flexible circuits. The quality of the printing layers greatly depends on the flatness and uniformity of the fabric. In this paper, a lightweight nylon fabric with a 0.1 mm thickness and warp and weft density of 68D and 100D was chosen.

The conductive ink was composed of nano-silver, binder, solvent, and other functional additives. The content of nano-silver was 75 ± 2%, which was purchased from Shenzhen Sunflower Electronic Material Co., Ltd. (Shenzhen, China). As shown in Table 1, it has a substantial solids content and appropriate viscosity, which ensure excellent electrical conductivity and stability for the printing layers. After removing the conductive ink from the refrigerator, it was allowed to stay at room temperature (approximately 25 °C) for more than an hour before being agitated at a low speed for 15 min to ensure the ink was fully mixed before printing. After repeated attempts, the best conductivity was achieved by drying the samples in a blast oven at 120 °C for 15 min.

### 2.2. Design and Fabrication

The function of the ECG machine is to amplify the weak electrophysiological signals and filter environmental and internal noise. In this paper, the fabric-based circuit is a biopotential amplifier circuit which aims to replace the key function of an ECG machine. The circuit was designed by utilizing the classic software of Altium Designer (version number: 10.1377.27009). Since the ECG signals are very weak, typically at the millivolt level, the amplification of the circuit will be designed to be over 1000 times. As the noise signals are amplified together with the electrophysiological signals, a multi-stage filter circuit must be designed to eliminate or avoid the artifacts such as 50 Hz pick-up, muscle shaking, and baseline or DC drift. A high input impedance and high common-mode rejection ratio (CMRR) are two necessary properties of a front amplifier to reduce significant offset signals [25,26]. To improve the signal integrity and reduce noise from the inputs, a classic circuit of a differential amplifier was adopted [27]. All in all, the realization of the above functions is dependent on the circuit’s rational design and the proper selection of component characteristics like resistors, inductors, capacitors, op-amps amplifiers, and so on. Particularly, the circuit function is accomplished by flexographic printing technology on flexible textiles, and unlike standard PCB chemical etching, the circuit is designed to planarize multi-layer circuits, which must avoid through-holes, via-holes, and other issues.

As shown in Figure 2, the biopotential amplifier circuit was eventually designed to comprise five modules including a front amplifier, second amplifier, electromyographic filter, 50 Hz notch filter, and level-up circuit [26]. It is expected to achieve 1600 times the signal amplification while also effectively addressing the 50 Hz interference, baseline drift, muscle shaking, and other interference issues in the process of electrophysiological signal acquisition and processing, making it suitable for the real-time monitoring of signals such as ECG, EMG, and joint movements.

The manufacturing process of the fabric-based circuit is illustrated in Figure 3. Initially, to improve the printing uniformity, the fabric is initially treated to cover a calcium carbonate, which provides a smooth and flat surface to realize a good printing quality. Next, the conductive ink is extruded by a microelectronic printer (MP1200, Shanghai Mifang Electronic Technology Co., Ltd., Shanghai, China) onto the fabric in accordance with a predesigned circuit. Following numerous testing and careful evaluation of the ink viscosity, printing stability, and accuracy, the final selection of printing parameters is shown in Table 2. In addition, the materials of the conductive ink and fabric substrate as well as the curing conditions were selected to ensure that the printed circuits fulfill the requirements of an excellent electrical performance and stability. The circuit topology is designed to avoid crossings between the ground and top layers as much as possible. It is notable that for the crossing component, an insulating film is printed locally at the crossing point for isolation treatment before connecting the top layer. And then, the printing process was repeated to complete the second layer of conductive lines to form the ground structures.

The electronic components (chips, resistors, capacitors, etc.) were subsequently firmly adhered to the fabric-based circuit by dripping a layer of transparent adhesive on them and fixed by the hot and press encapsulation process. Lastly, the pin section of the electrical component was partially linked with conductive ink and dried once again in an oven to further improve the conductivity. Before testing, the conductivity of all component pins and wires of the circuit was checked with a multimeter. The entire circuit was encapsulated with a layer of elastic film for further protection after the chip and component packaging was finished and the basic conductivity was verified.

### 2.3. Microscopic/Epidermal Morphological Characterization

In order to study the performance of printed conductive circuits, the surface/ cross-section morphology and elemental distribution were analyzed by scanning electron microscopy (Regulus 8100, Hitachi High-Tech Corporation, Tokyo, Japan), and the results are shown in Figure 4. As illustrated by the picture, the silver nanoflakes were homogenously distributed in the surface of printed circuit, without obvious splitting or clustering. By macroscopic inspection of its surface (magnified 50 times) and cross-section (magnified 200 times), it can be concluded that the line width is 0.9 mm and the thickness of the conductive layer is about 20–30 μm. In addition, the element mapping and the energy-dispersive spectrometry (EDS) spectra are shown in Figure 4e,f. In Figure 4e, the yellow particles represent the carbon element (C), and the pink particles represent the silver element (Ag). It can be seen that the silver elements are completely and evenly distributed in the printed trace. Figure 4f reveals that the content of silver nanoflakes of the printed circuit after curing is 54.56 wt%, which ensured the good electrical conductivity of the circuit.

The printing accuracy of the circuit, line width, and spacing must be controlled in a specified range. Otherwise, it is easy to produce circuit breakage, parallel neighboring circuits, or component pin misalignment, and other issues. Therefore, it is necessary to evaluate the geometrical properties of the printing circuits.

As shown in Figure 5a, the width and inhomogeneity of the printing layers were measured by an optical microscopy, and the average width is 0.90 ± 0.10 mm, which is consistent with the surface morphology by SEM (shown in Figure 4c). On the other hand, the thickness of the printed conductive layers was measured by the level meter (KLA, Alpha-Step D-300), and the result shows that the average thickness is 25.19 ± 8.76 nm (shown in Figure 5b) which is also consistent with the cross-sectional thickness measured by SEM (shown in Figure 4d). According to the scanning results of the step meter, the thickness of the printed film is thinner on both sides and thicker in the center, which is connected to the ink viscosity, printing process parameters, curing conditions, and other factors.

### 2.4. Electrical/Electrochemical Characterization

In order to further investigate the durability of printed fabric-based circuits, the electrical properties after cyclic stretching and bending were investigated. The cyclic tensile/flexural test was performed on a flexible electronic tester (FT2000, Shanghai Mifang Electronic Technology Co., Ltd., Shanghai, China). In order to imitate the strain fatigue process of human’s daily activities, the test methods and procedures for tensile and flexural strain reference Yao et al. [28], which causes cyclic fatigue damage by controlling tensile displacement and the bending angle.

For the tensile test, the total length of the sample was set to be 5 cm, and the starting position of the stretching was 0 cm, while the ending position of the stretching was 5.1 cm. That is to say, the total stretching rate of the sample was controlled to be 2%. As displayed in Figure 6a, after 10,000 stretching cycles, the resistance of the sample remained mostly the same. Within the length of 6 cm, the total resistance rose slightly from 1.33 Ω to 1.35 Ω, and the average resistance remained at 0.27 Ω/cm after 10,000 cycles of stretching, indicating the exceptional electrical stability of the printed fabric-based circuits.

For the bending test, a sample with a length of 5 cm is bent repeatedly by swinging the rotor shaft. The bending angle and speed were set to be 30 degrees and 30 degrees/second, respectively. As displayed in Figure 6b, after 10,000 times of cyclic bending, the sample resistance rises slightly from 1.3 Ω to 3.2 Ω, and the resistance per unit of length rises from 0.26 Ω/cm to 0.64 Ω/cm, with an almost 150% increase in resistivity, which is much more mechanically durable than fabric-based graphene oxide circuits [8]. In comparison to the resistance of the circuit components, which is in the kilo-ohm range, the resistance change during the bending is minimal and can be ignored [23]. Therefore, the above two sets of cyclic tensile and flexural tests are sufficient to demonstrate the electrical stability of the circuit, which is of great significance for the application of this circuit in wearable electrophysiological monitoring.

## 3. Results and Discussion

### 3.1. ECG Signal Acquisition

In order to verify the function of the printed fabric-based circuit, it was firstly simulated and measured to compare with the design objectives. The simulation software is named Multisim (version 14.0), which was developed by National Instruments (NI), designed for analog and digital circuits’ design. As the minimum input amplitude of a signal generator is 4 mV, the input signal of the simulated software and the signal generator were both set to be the same. Here, the input signal set in the simulation and test was a sine wave with an amplitude of 4 mV and a frequency of 10 Hz. As shown in Figure 7, by observing the output waveforms of the simulation and experiment we may determine whether the circuit is working properly.

The simulation results show that the circuit outputted a sine wave with an amplitude of 6.68 V and frequency of 10 Hz, which indicated that the simulated amplification was 1670 times and there was no waveform distortion. On the other hand, the experimental result showed that the output signal was a sine wave with an amplitude of 6.72 V and frequency of 9.98 Hz. Here, the phase of the output signals of the simulation and experiment were in alignment with their input waveforms, respectively. The results of Figure 7b are only shown for the purpose of increasing the contrast effect, and they do not mean that the phase difference between the simulated and tested waveforms is 180 degrees. Finally, the experimental results show that the actual amplification is 1680 times. Both are closer to the designed amplification of 1600 times. The amplification error is about 5%, and the frequency error is within 0.2%, successfully verifying the function of the circuit. In terms of signal amplitude accuracy, the precision basically reaches the level of portable out-of-hospital electrocardiography [29]. The possible errors may be resistance of the printed wire, the error of the component resistance, the error of the measuring instrument, and so on.

Furthermore, the conventional signal generator was replaced by the ECG simulator in order to verify its amplification of the ECG signal [30]. The experimental connection is shown in Figure 8a and the results are shown in Figure 8b. The ECG signal simulator (KS-300, Shanghai Pukang Co., LTD, Shanghai, China) was set at a normal sinus rhythm with a heart rate of 60 BPM and a signal amplitude of 1 mV. The results showed that the printed fabric-based hybrid circuit can effectively detect and amplify the ECG signals, which provided a solid foundation for detecting and amplifying real human electrophysiological signals.

### 3.2. Real-Time Electrophysiological Signal Monitoring

For a potential smart clothing or functional textile application, the capacity to monitor humans’ electrophysiological signals in real time is critical. Finally, a series of electrophysiological signals from the human body, such as ECG, EMG, and joint movement signals, were monitored by this printed fabric-based circuit. The physical diagram of the fabric-based hybrid electric circuit, the schematic diagram of signal monitoring, and the functional blocks of the designed circuits are shown in Figure 9a–c, while the experimental results are shown in Figure 9d–f.

It is crucial to note that the aim of this printed fabric-based hybrid electric circuit is to replace the key function of an ECG or EMG machine. The circuit is flexible and stable enough to be worn on any area of the body or incorporated into clothing, and the joints should be avoided as much as possible. The electrical activity of the point can be tested by simply placing the electrode at the appropriate location. In the future, the multifunction of sensing, storing, and computing can be realized by integrating measurement electrodes with this key circuit through preparation in the same preparation process in one suit. The subject is a normal female college student who is 21 years old with a height of 160 cm and a weight of 50 Kg. The process of ECG signal monitoring is shown in Figure 9a, and the physical diagram of the fabric-based circuit is shown in Figure 9b. Figure 9c illustrates the operation principles of an ECG machine.

The ECG signal obtained from the test is shown in Figure 9d. There are significant QRS wave features and the noise and interference have largely been eliminated with a more stable baseline. By calculation, the time interval between the two adjacent QRS wave peaks is 0.83 s, which means the heart rate is approximately 72 BPM. The maximum amplitude of the amplified QRS wave peaks presented is approximately 1.2 V. Overall, the characteristics of the ECG signals are very obvious, indicating that the circuit successfully realizes the real-time monitoring of ECG signals, including the functions of amplification and filtering ECG signals [19].

Next, the electrodes were placed on the back of the hand for EMG signal monitoring and amplification. When the palm was regularly stretched and clenched into a fist, the acquired EMG signals can be seen in Figure 9e. The measured EMG signals almost jumped up and down at 0 V when the palm was almost static and relaxed, and the amplitude of the EMG signals increased rapidly to more than 100 V when the palm was quickly clenched into a fist. The actual EMG signals are different because the force applied to the human hand is not exactly the same for each stretching and clenching of the hand, but they do provide a more complete representation of muscle activity variances under various stress situations [31].

Finally, the electrodes were placed on the knee joint to monitor and amplify joint activity, and the acquired signals are shown in Figure 9f. The cyclic process of slowly raising the knee joint to a certain angle and returning to the original position showed a certain stable action potential change with an amplitude of about 100 V. In fact, the electrical signals generated by the joint movement are also a manifestation of electromyographic signals. The above results demonstrate that the printed fabric circuit properly realizes the function of the real-time monitoring of electrophysiological signals, hence validating the circuit’s successful preparation.

## 4. Conclusions

This study presented a significant advancement in the development of smart clothing for health monitoring. By creating a printed fabric-based hybrid electric circuit, the research overcame the limitations of bulkiness and inconvenience associated with current wearable health monitoring devices. The experimental results demonstrated the prominent conductivity and reliability of the printed fabric-based circuits, even under repeated stretching and bending cycles. Moreover, the integration realized the function of 1600-times amplification and various filtering, efficiently capturing and processing weak electrophysiological signals such as ECG, EMG, and joint motions. It provided valuable and practical guidance for the development of unobtrusive and real-time health monitoring for smart clothing.

## Figures and Tables

**Figure 1 sensors-24-06991-f001:**
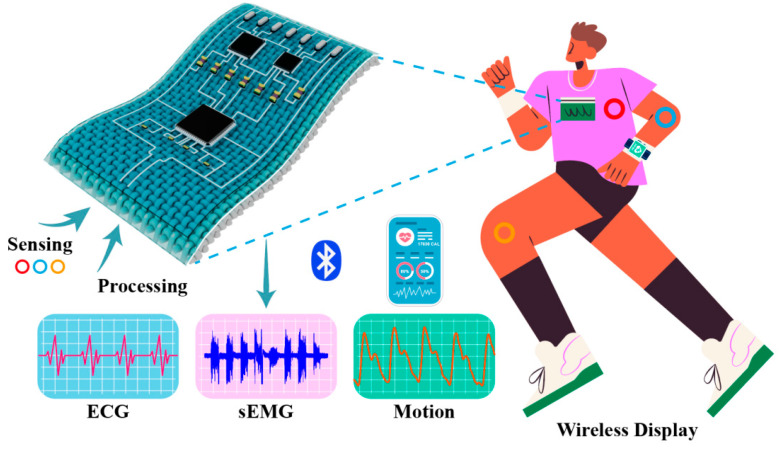
A fabric-based hybrid circuit for wearable health monitoring.

**Figure 2 sensors-24-06991-f002:**
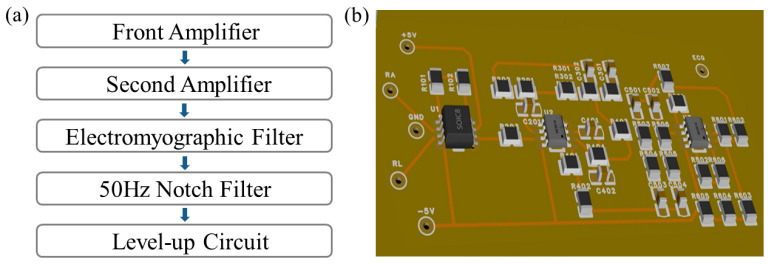
The biopotential amplifier circuits: (**a**) the framework diagram of the designed circuit; (**b**) the layout of the designed circuit.

**Figure 3 sensors-24-06991-f003:**
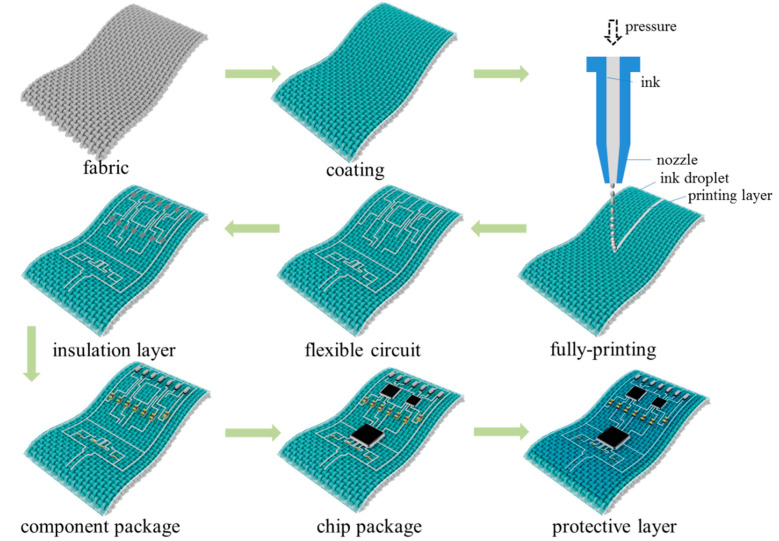
The manufacturing process of the fabric-based hybrid electric circuit.

**Figure 4 sensors-24-06991-f004:**
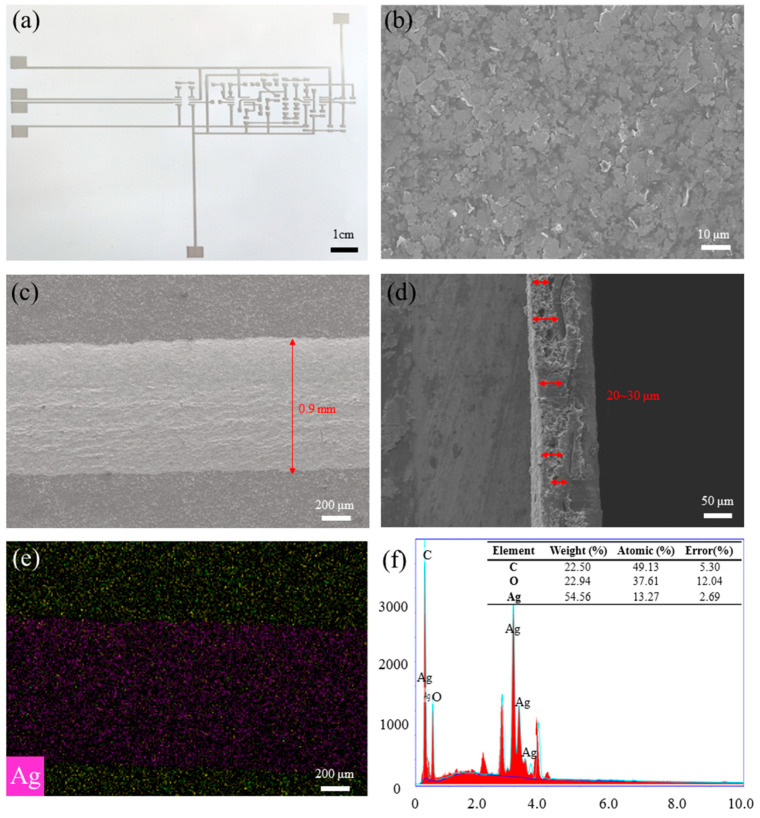
The surface and cross-section morphology of the printing circuit: (**a**) the overall shape; (**b**) the surface morphology with a magnification of 1000 times; (**c**) the width of the conductive line is about 0.9 mm with 50 times magnification; (**d**) the thickness of the conductive line is about 20~30 μm (red arrow area) with a cross-sectional magnification of 200 times; (**e**) element mapping; (**f**) the EDS spectra.

**Figure 5 sensors-24-06991-f005:**
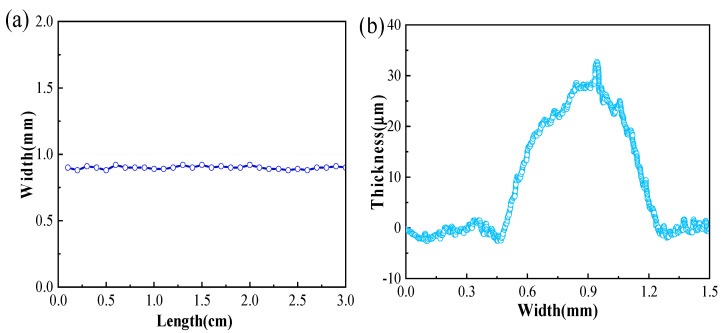
Geometrical properties of printing circuits: (**a**) the thickness uniformity; (**b**) the line width uniformity (the pink color means the width/thickness area of printed lines).

**Figure 6 sensors-24-06991-f006:**
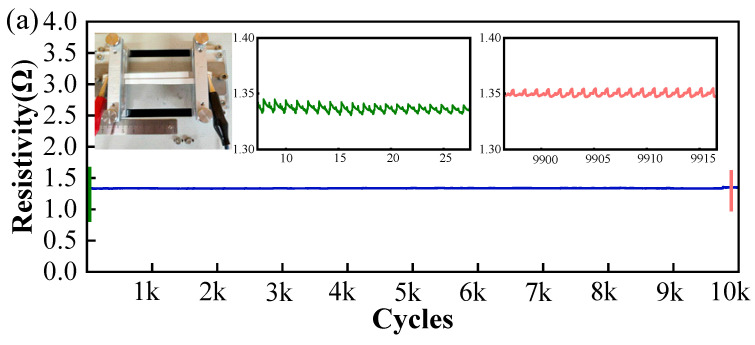
Resistance shift after 10,000 cycles of tensile and flexural conditions: (**a**) tensile test (the blue line means the resistance shift during 0 to 10,000 cycles of stretching; the green and red lines are localized zoomed-in plots, indicating the changes in resistance during the initial and final time periods, respectively); (**b**) bending test (the blue line means the resistance shift during 0 to 10,000 cycles of bending).

**Figure 7 sensors-24-06991-f007:**
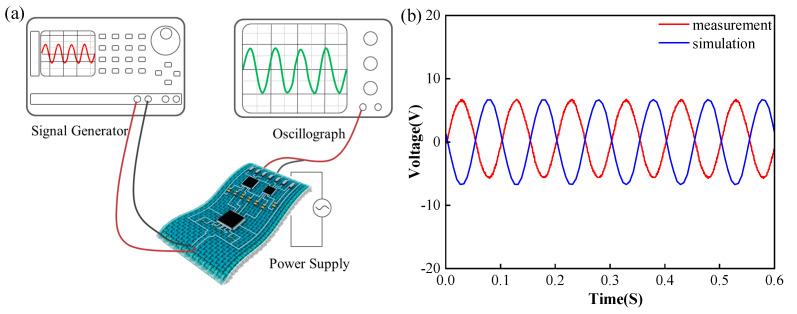
The functional verification of the printed fabric-based circuit: (**a**) schematic diagram of the measurement process; (**b**) comparison of the simulation and measurement results with a sine wave.

**Figure 8 sensors-24-06991-f008:**
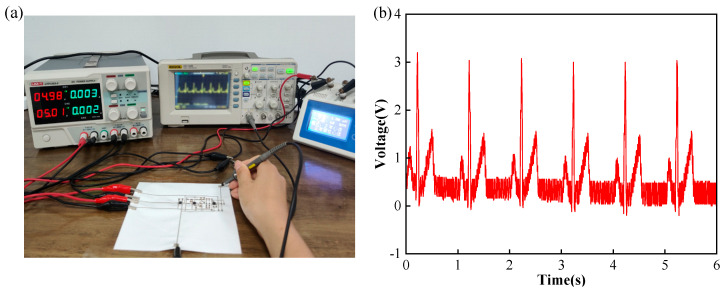
ECG signal measurement: (**a**) physical diagram of the measurement process; (**b**) measurement results with the ECG signal generator.

**Figure 9 sensors-24-06991-f009:**
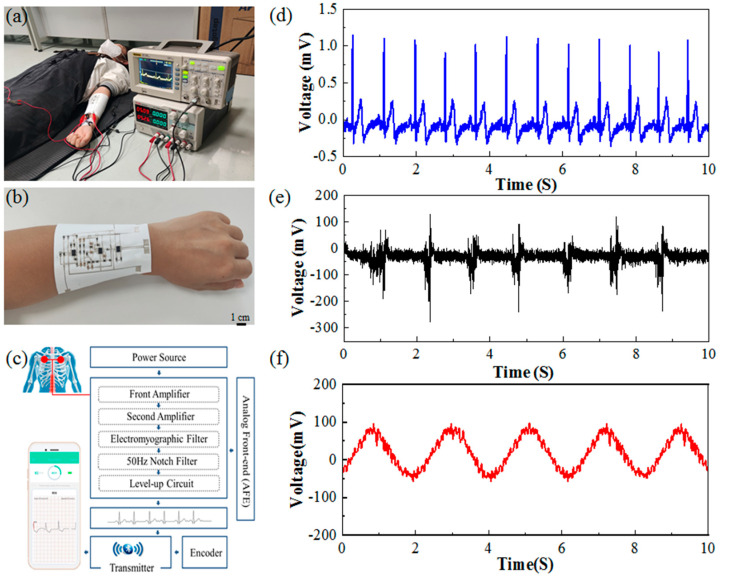
Real-time electrophysiological signal monitoring on humans: (**a**) process of ECG signal monitoring; (**b**) physical diagram of a fabric-based circuit; (**c**) schematic diagram of the designed circuit; (**d**) results of ECG signal monitoring; (**e**) results of EMG signal monitoring; (**f**) results of joint motion signal monitoring.

**Table 1 sensors-24-06991-t001:** The performance parameters of conductive ink.

Title	Model	Viscosity	Ag Content	Drying Condition	Manufacturer
Conductive ink	8000C	10,000–12,000 cp	75 ± 2%	hot air at 120 °C, 15 min	Shenzhen Sunflower Electronic Material Co., Ltd. (Shenzhen, China).

**Table 2 sensors-24-06991-t002:** The printing parameters of the fabric-based hybrid electric circuit.

Inkjet Height	Inkjet Speed	Dispensing Air Pressure	Needle Diameter
31.90 mm	2.00 mm/s	200.00 KPa	0.25 mm

## Data Availability

The raw data supporting the conclusions of this article will be made available by the authors on request.

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
