# Peer review of "A Step Forward for Smart Clothes: Printed Fabric-Based Hybrid Electronics for Wearable Health Monitoring"

_sensors, 2024, doi:10.3390/s24216991_

Round 1

Reviewer 1 Report

Comments and Suggestions for Authors

The author should add the information about conductive ink..." the conductive ink is based on ..." Are nickel, carbon or graphene particles? 

Why it is necessary to keep the conductive ink in the refrigerator (which temperature was used?)?

This conductive ink is resistant to washing and detergents.

In Figure 4c, what do the yellow and pink particles represent?

In Figure 8, is the support for the printing circuit a textile material or just simple paper?

These printed circuits have good resistance to abrasion, bending or traction.

Author Response

Comments 1: The author should add the information about conductive ink..." the conductive ink is based on ..." Are nickel, carbon or graphene particles?

Response 1: Thank you for making this valuable suggestion. The conductive ink was composed of nano-silver, binder, solvent and other functional additives. The conductive ink is based on silver, which is the most dominant component and determines the electrical properties of the printed layer.

It has been replenished on page 3 of the revised and marked manuscript.

Comments 2: Why it is necessary to keep the conductive ink in the refrigerator (which temperature was used?)? This conductive ink is resistant to washing and detergents.

Response 2: Thank you for pointing this out. We agree with this comment. The primary ingredients of conductive silver ink are nano-silver, solvent, binder, and additional useful additions. The viscosity and surface tension of silver conductive inks are maintained within an acceptable range for inkjet printing by storing them in a refrigerator, which also slows down the evaporation of solvents and the breakdown of organic components in the ink. It is suggested that silver conductive inks be kept in a refrigerator for best results and longer shelf life. By doing this, the ink's chemical and physical characteristics are preserved before printing, preventing variations in room temperature from impairing the ink's printing capabilities. Regular refrigerator temperatures range from 4 to 8℃.

Comments 3: In Figure 4e, what do the yellow and pink particles represent?

Response 3: Thank you for making this valuable suggestion. The main components of conductive silver ink are nano- silver, binder, solvent and additives to improve performance. The main elements are Ag, C. O .N, some may also have special trace elements. Figure 4e is the element mapping of the printed conductive layer, and the yellow particles represent Carbon element(C), and the pink particles represent Silver element(Ag). It has been replenished on page 6 of the revised and marked manuscript.

Comments 4: In Figure 8, is the support for the printing circuit a textile material or just simple paper? These printed circuits have good resistance to abrasion, bending or traction.

Response 4: Thank you for your valuable and thoughtful comments. The flexible hybrid circuits fabricated in this manuscript are created by printing conductive ink on fabric, which is composed of textile materials of nylon with calcium carbonate coating, which ensures a smooth and flat surface to realize good properties such as printing quality, resistance to abrasion, bending, or traction. The material has a smooth, white appearance and appears to be regular paper, yet it is not it is made of textiles.

Reviewer 2 Report

Comments and Suggestions for Authors

The paper is well written, and I agree with the publication. There are some editing mistakes in the text but I think they will be corrected by the journal' editing team.

Author Response

Comments 1: The paper is well written, and I agree with the publication. There are some editing mistakes in the text but I think they will be corrected by the journal' editing team.

Response: Thanks for your review and valuable comments.

Reviewer 3 Report

Comments and Suggestions for Authors

Comments to the Authors

This study introduces an innovative approach to the development of smart clothing for real-time health monitoring by using a printed fabric-based hybrid circuit. While traditional health monitoring devices often rely on conventional printed circuit boards (PCBs), making them bulky and uncomfortable, the use of flexible fabric circuits addresses this issue, allowing for more comfortable and wearable technology.

Key aspects of the research include: (1) Fabric-based Hybrid Circuit: The hybrid circuit is printed on fabric, offering greater flexibility and comfort compared to rigid PCBs. (2) Characterization: The fabric circuit underwent extensive testing, including surface and sectional morphology, electrical properties, and stability under physical stress, to ensure reliability. (3) Mechanical Durability: The circuit demonstrated excellent durability, maintaining conductivity after 10,000 cycles of stretching and bending, with resistance remaining stable at 0.27 Ω/cm for elongation and 0.64 Ω/cm for bending. (4) Signal Processing: The fabric-based circuit successfully amplified weak electrophysiological signals, such as ECG signals, by 1600 times, while also filtering out environmental noise and industrial frequency interference (50 Hz). (5) Applications: The fabric circuit was effective in monitoring various electrophysiological signals, including ECG, EMG, and joint movements, making it highly suitable for unobtrusive health monitoring in smart clothing.

Therefore, this study can be recommended for publication after the following comments are reflected in the revision.

1.      In abstract and figure 9, the authors claim that the circuit can successfully amplify the weak ECG signal and the magnification is 1600 times with filtering of environmental and 50 Hz industrial frequency interference. I strongly recommend providing animation data of the wireless real-time monitoring of ECG, EMG, and joint motion signals to support the claims.

2.      In page 10, the authors claim that “The subject is a normal female college student who is 21 years old with a height of 160 cm, and weight of 50 Kg. The ECG signal obtained from the test was shown in Figure 8(d).” However, it lacks information such as where the electrodes are attached. The authors should be added.

3.      In abstract, the authors do not provide the definition of ECG and EMG (first time used).

4.      In page 2, the authors do not provide the definition of PET, PCB, and FPCB (first time used).

5.      In page 3, “fifteen minutes” should be “15 minutes”

6.      In page 4, “electrocardiograms, electromyograms,” should be “ECG, EMG,”

6.

7.      In page 5, the authors say “Initially, to improve the printing uniformity, the fabric is initially treated to cover a coating.” The procedure is poorly described. For example, coating material?

8.      In page 5, “ Figure 4(d) and (e)” should be “Figure 4(e) and (f)”

9.      In page 8, “ Figure 6” should be “Figure 7”

10.  In caption of Figure 9(e), “myoelectric signal monitoring” should be “EMG signal monitoring”

Comments on the Quality of English Language

Moderate editing of English language required.

Round 2

Reviewer 3 Report

Comments and Suggestions for Authors

It is quite clear that there is a massive work done. The fabrication method is well explained, and all the relevant details are reported in the manuscript.